# Chronic Back Condition and the Level of Physical Activity as Well as Internet Addiction among Physiotherapy Students during the COVID-19 Pandemic in Poland

**DOI:** 10.3390/ijerph18136718

**Published:** 2021-06-22

**Authors:** Monika Gałczyk, Anna Zalewska, Irena Białokoz-Kalinowska, Marek Sobolewski

**Affiliations:** 1 Faculty of Health Sciences, Lomza State University of Applied Sciences, Akademicka 14, 18-400 Lomza, Poland; aanna.zalewska@gmail.com (A.Z.); ibialokozkalinowska@pwsip.edu.pl (I.B.-K.); 2 Plant of Quantitative Methods, Rzeszów University of Technology, al. Powstancow Warszawy 12, 35-959 Rzeszow, Poland; mareksobol@poczta.onet.pl

**Keywords:** ODI, NDI, COVID-19, IPAQ, Kimberly Young Questionnaire

## Abstract

**Objectives:** The aim of this study was to assess back pain and its relation to physical activity as well as Internet addiction among Polish university students during the COVID-19 pandemic. **Methods****:** The research was conducted via the Internet in student groups of three universities in Poland (141 people). Back pain was examined by ODI—The Oswestry Disability Index and NDI—Neck Disability Index. The Polish-language International Physical Activity Questionnaire (IPAQ) was used to assess physical activity and the level of Internet addiction was tested using the Kimberly Young Questionnaire. **Results****:** The respondents mostly suffered from pain originating in the cervical spine. In the study group, only every fifth person had a high level of physical activity. Sex differentiates the level of the measures considered. Sitting in front of the computer affects the pain in the spine. **Conclusions:** Research results suggest that the pandemic is negatively affecting students. Frequent occurrence of back pain is observed with a simultaneous low level of physical activity. Maintaining regular activity during a pandemic, at least at home, is an indispensable preventive measure for physical health.

## 1. Introduction

The COVID-19 pandemic has caused numerous changes worldwide in many areas of socio-economic life. Several restrictions changing the lifestyle of Polish students were introduced as a result of the announcement of the epidemic in Poland. The introduced restrictions forced a change in the nature of science; the form of full time classes were replaced with online study. Considering this type of education, it is difficult to imagine functioning without access to the Internet. The period of the pandemic contributed even more to regular online presence, primarily for teaching and educational purposes. Excessive use of the Internet is associated with several health problems, such as the possibility of developing musculoskeletal pain or a decrease in the level of physical activity [1,2].

The lockdown fostered progressive change of lifestyle from being physically active to passive and sedentary, leading to a reduction in the spine resistance to static-dynamic loads, which in turn may lead to pain in the spine [3,4].

Spinal pain in the contemporary world is a growing problem that threatens physical health, as well as reduces the quality of work and everyday life [5,6].

Collective isolation lasting for several months may lead to a limitation of the current physical activity among students [7]. Daily physical activity at a low or moderate level may be treated as one of the factors which positively influence physical and mental health. Physical benefits include, for example, maintaining body weight at an appropriate level or lowering blood pressure [8]. Considering mental health, physical activity improves the functioning of the brain and helps to achieve better results in learning [9].

The aim of the study was to assess back pain and its relation to physical activity as well as Internet addiction among Polish university students during the COVID-19 pandemic.

## 2. Materials and Methods

### 2.1. Participants

A cross-sectional survey was sent to physiotherapy students of four universities in Poland (Lomza State University of Applied Sciences, Medical University of Bialystok, School of Medical Science in Bialystok, Medical University of Warsaw) during the second COVID-19 lockdown in November 2020 (after 9 months of online education due to the pandemic). The survey was conducted on an official educational platform. Students were given a link to the Google form with the survey, as well as information about the survey, its purpose and the anonymity of participants. They also had to sign an informed consent for participation in a survey. This method was chosen because of epidemiological safety reasons and in order to obtain a prompt response. 141 people (104 women and 37 men) aged 18–25 took part in the study. This population was considered homogeneous in terms of age and this factor was not taken into account in further analyses. The study group is a representative sample for the population of physiotherapy students in Poland aged 18–25.

The following inclusion criteria were used: (a) age >18, (b) the status of a physiotherapy student, (c) correct completion of the questionnaire, (d) informed consent for participation in the research.

The research was approved by the Senate Commission for Ethics in Scientific Research of the School of Medical Science in Bialystok KB/18/2020.2021 and the informed consent of the participants was obtained. Participation in the study was voluntary, and its course was based on the Personal Data Protection Act of May 10, 2018 (Journal of Laws of 2018, item 1000) in accordance with the Regulation of the European Parliament and the Council (European Union) 2016/679 as of 27 April 2016, on the protection of individuals concerning the processing of personal data and on the free movement of such data, and the repeal of Directive 95/46/WE (General Data Protection Regulation). Written consent was obtained from the participants.

On the basis of the standards created for the IPAQ questionnaire, the respondents were categorized into three groups of low (40.4%), medium (39.7%) and high (19.9%) physical activity. Due to the high influence of gender on the analyzed measures (especially activity measures), the analyses were performed with the subdivision into the groups of women and men.

### 2.2. Methods for Assessing Physical Activity, Spinal Pain, Internet Addiction

The studies presented were self-managed cross-sectional studies. The questionnaires were distributed by the authors of the article via the Internet in student groups of three universities in Poland.

Spinal pain was assessed by the Oswestry Disability Index/NDI—Neck Disability Index (ODI). The Oswestry Questionnaire is a reliable, widely used worldwide criterion for assessing the disability of people with back pain syndrome. It includes 10 questions concerning important activities of everyday life, such as pain intensity, self-service, lifting, walking, sitting, standing, sleeping, social life, traveling and work. Each answer is scored a: answer A—0 points, answer B—1 point, answer C—2 points, answer D—3 points, answer E—4 points and answer F—5 points. The points are then added up; the maximum number of points is 50.

Disability rating scale:0–4: none5–14: small15–24: mediocre25–34: seriousover 35: total

The short version of the Polish-language International Physical Activity Questionnaire (IPAQon), intended for people aged 15–69, was used to assess physical activity. The survey consists of 7 questions concerning all types of physical activity related to everyday life. The questionnaire takes into account activities that last continuously for at least 10 min. The different types of physical activity are expressed in units of MET—min/week, by multiplying the coefficient assigned to a given activity by the number of days it is performed per week and the duration in minutes per day. Intensity coefficients, corresponding to the multiple of the basic transformation (Metabolic Equivalent of Work—MET), are used to assess specific types of activity.

The level of Internet addiction was examined with the use of the Kimberly Young Questionnaire, which consists of 20 questions on various aspects of the frequency of Internet use. The respondents are to provide answers on a 5-point scale (1—rarely, to 5—always). The summary measure ranges from 20 to 100 points, with higher values indicating greater dependence on the Internet. This measure is categorized into three groups of low, medium and high Internet addiction (for point ranges: 20–49, 50–79 and 80–100, respectively).

### 2.3. Statistical Methods

The distribution of both the values of psychometric measures and measures of activity was characterized by selected descriptive statistics: the mean with the 95% confidence interval, the median, the standard deviation, and the skewness coefficient. The latter measure was significantly different from the zero value, meaning a symmetric distribution, and for several measures its value was significantly above 1, which means a clear right-hand asymmetry. This fact forced the choice of non-parametric statistical inference methods, such as the Mann–Whitney test, Kruskal–Wallis test or the Spearman’s rank correlation coefficient.

## 3. Results

It was more often and more severe for the respondents to suffer from pain in the area of cervical spine (Table 1). All descriptive parameters related to this segment were higher than those of the lumbosacral area.

The chart (Figure 1) shows the percentage distribution of the values of both measures in intervals of 5 points. The asymmetry in the distribution of ODI (The Oswestry Disability Index) and NDI (Neck Disability Index) measures is related the fact that young people having no or very little back pain problems were assessed in the study. 

On the basis of the information obtained with the use of the short version of the IPAQ questionnaire physical activity (IPAQ-International Physical Activity Questionnaire) was measured in the sphere of high-intensity, moderate and light activity (Table 2). Measurements of activity are usually very asymmetric, which results from the presence of a relatively small group of people with a very high level of activity, as well as a fairly large group of people characterized with 0 activity. Therefore, a better measure of the average level is the median than the mean, which is overestimated by the above-described positive outliers: The measure of vigorous activity is slightly higher than that of walking (median values of 320 and 297 MET-min/week, respectively), while the measure of moderate intensity activity has a much lower value (median of 160 MET-min/week). week).

In the studied group, the situation was not good, because less than every fifth person could be distinguished by a high level of physical activity.

Gender differentiates the level of the measures considered, with the exception of low-intensity activity (walking) (Table 3). Men are likely to present with significantly lower levels of pain (especially ODI), are more addicted to the Internet and show a much higher level of physical activity.

The correlation analysis was carried out according to the formulated research problems (Table 4). The Spearman’s rank correlation coefficient with the assessment of statistical significance (*p* < 0.05) was shown in the tables. The table shows the Spearman’s rank correlation coefficients between the severity of pain and physical activity (IPAQ measures). No significant correlations were observed in the group of women, while among men there is a statistically significant (for the measure of total activity) and close to statistically significant (for the partial measures of activity) relationship with NDI. The higher the pain in the cervical area, the slightly lower the physical activity (the correlations are not strong; the absolute value of rS is approximately 0.30).

The graphs (Figure 2) show the results of the analyses concerning the measure of total activity as well as NDI and ODI pain.

Sitting in front of a computer or smartphone influences back pain (Table 5). Such a relationship occurs mainly as for cervical pain (correlations are statistically significant (*p* < 0.05) for both sexes, but the correlation strength is greater for men: rS = 0.37 vs. rS = 0.20), and in the case of female students, on the border of statistical significance there is the relationship between the Internet addiction and pain in the lumbosacral region (rS = 0.19; *p* = 0.0589).

## 4. Discussion

Online electronic questionnaires are an easy to implement and safe (especially during an epidemic) method of collecting information [10]. The period of the COVID-19 pandemic, as well as the associated isolation and online learning, had a negative impact on many areas of life.

In response to the spreading pandemic, the government significantly limited all motor behavior and enforced social distancing from citizens. Restrictions were placed on the activities of fitness industry and swimming pools. Forests, boulevards and parks were closed during a certain period of the pandemic. People could not leave their places of residence, and so learning at universities was conducted online.

Pain in the cervical and lumbar area of the spine may occur due to certain factors such as stress and psychosocial factors [11]. The high risk of developing musculoskeletal pain [12,13,14] appears more among students of medical professions than in other populations.

Our research assessed young people learning online amid realities of the pandemic. A statistically significant relationship was found between Internet addiction and lumbosacral pain. Similar relationships have been shown in previous studies in which the excessive use of electronic devices was a risk factor for the development of cervical and lumbar spine pain [12,15,16].

Interestingly enough, in the authors’ own research, the respondents mostly suffered from pain originating in the cervical spine. During the pandemic and contact restrictions, social life and science moved online. Even before this phenomenon, a positive correlation was observed between the time of using a mobile phone and the duration and intensity of neck pain [17]. These ailments could be reduced by introducing ergonomic measures [18].

Our research also shows that more intense pain in the cervical spine was positively correlated with lower physical activity, which is confirmed by the results of studies conducted among students before the pandemic [19]. Other authors also refer to the relationship between back pain and lack of physical activity [20,21].

In terms of physical activity, the situation is not good in the studied group of physiotherapy students. Almost in every fifth person the level of physical activity was estimated as high. Similar results regarding the limitation and low level of physical activity were noticed among students during the pandemic in England [22], Scotland [23], Italy [24], Spain [25], Switzerland [26], Hungary [27], USA [28], Mexico [29] and China [30]. A decrease in the level of physical activity among students of physiotherapy was observed in India, where its level was defined as medium [31]. The authors’ own research, conducted among Polish students, shows a very disturbing level of activity—low in 40.4% of the respondents. Similar results were obtained among Italian students—39.62% of them showed a low level of physical activity during the epidemic [32]. These results are alarming in the context of global studies conducted before the pandemic, which already estimated that 27.5% of the population had insufficient levels of physical activity [33].

A decrease in physical activity was more frequently observed in women. Other results were obtained in Great Britain [22], Spain [34] and Italy [35]. Savage et al. and Maugeri et al. suggest that such a situation may be caused by the preference for sports among men for social and competitive reasons as well as in public places such as gyms and fitness clubs (which was prevented by the development of the pandemic), while women (who are more likely to exercise at home) practice sports for reasons related to maintaining an adequate body weight [22,32]. The discrepancy in the results may be related to a small study group of men.

Prolonged periods of limitation in physical activity, sedentary lifestyle and isolation have a negative impact on depression and the feeling of anxiety and fear [36]. It may also be associated with the decrease in immunity [37] and undertaking risky behaviors [37]. One should notice that regular physical activity strengthens the immune functions of the body, which can contribute to reducing the risk of infection. The negative effects of the COVID-19 pandemic, prolonged lockdown and restricted social contact may have consequences for years.

The limitations of our study were the small size of the group and the significant predominance of female students among respondents, as well as the subjectivity of respondents in the online self-report survey. Subsequent research should cover students of other faculties, as well as individual age or professional groups. Future research should be oriented towards a larger sample and use more objective methods to evaluate the parameters tested.

## 5. Conclusions

Research results suggest that the pandemic is negatively affecting students. Frequent occurrence of back pain is observed with a simultaneous low level of physical activity. Maintaining regular activity during a pandemic, at least at home, is an indispensable preventive measure for physical health. Research results should be taken into account by university authorities in order to organize long-term support for their students. Activities aimed at promoting physical activity can improve students’ physical health and help them to cope better in uncertain times. This study supplements the literature on back pain and physical activity during the COVID-19 pandemic.

## Figures and Tables

**Figure 1 ijerph-18-06718-f001:**
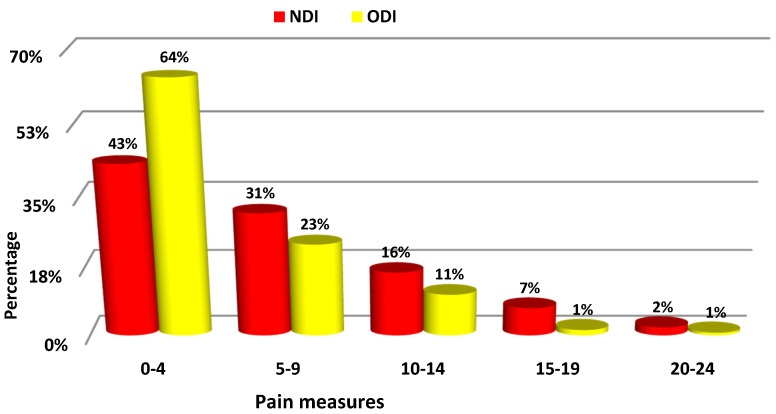
Percentage distribution of NDI and ODI measures.

**Figure 2 ijerph-18-06718-f002:**
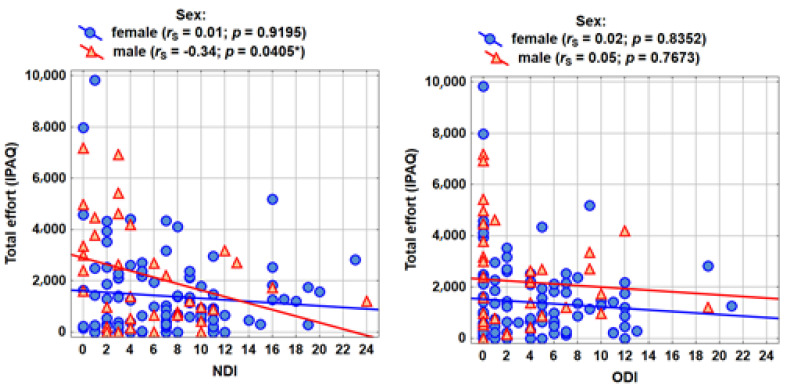
Measurements of total activity of pain NDI and ODI.

**Table 1 ijerph-18-06718-t001:** Back pain—descriptive parameters.

Back Pain	Mean	Median	Std. Dev.	Min	Max	Skewness
NDI (cervical)	6.6	6	5.1	0	24	1.01
ODI (lumbosacral)	3.9	2	4.5	0	21	1.31

**Table 2 ijerph-18-06718-t002:** Type of activity (IPAQ)—descriptive parameters.

Type of Activity (IPAQ)	Activity Performed during the Week (MET-min/week)					
	Mean	Median	Std. Dev.	Min	Max	Skewness
Intense effort	778	320	1145	0	6160	2.23
Moderate effort	338	160	513	0	3080	2.41
Walking	501	297	555	0	2541	1.49
Total effort	1617	1030	1760	0	9834	1.82

**Table 3 ijerph-18-06718-t003:** Gender and measures of pain, Internet addiction and activity.

Variable	Sex				*p*
	Female		Male		
	Mean (95% C.I.)	Median	Mean (95% C.I.)	Median	
NDI	7.0 (6.0–7.9)	7	5.4 (3.7–7.2)	4	0.0730
ODI	4.1 (3.3–5.0)	2.5	2.9 (1.5–4.4)	0	0.0400 *
Internet addiction	32.0 (30.2–33.8)	30	36.2 (32.3–40.0)	34	0.0495 *
Intense effort	625 (421–830)	160	1209 (778–1640)	880	0.0015 **
Moderate effort	259 (168–350)	80	559 (368–750)	320	0.0006 ***
walking	522 (409–636)	314	440 (283–597)	297	0.6225
Total effort	1407 (1091–1722)	900	2207 (1538–2877)	1596	0.0352 *

*p*—test probability value calculated using the Mann-Whitney test, statistically significant differences were denoted with *, ** or *** respectively for p <0.05, *p* < 0.01 or *p* < 0.001; NDI (Neck Disability Index), ODI (The Oswestry Disability Index).

**Table 4 ijerph-18-06718-t004:** Correlations between back pain (ODI and NDI) and the level of physical activity (IPAQ).

Level of Activity (IPAQ)	Sex			
	Female		Male	
	Spinal Pain			
	NDI	ODI	NDI	ODI
Intense effort	−0.13 (*p* = 0.1881)	−0.06 (*p* = 0.5777)	−0.31 (*p* = 0.0616)	0.01 (*p* = 0.9307)
Moderate effort	−0.15 (*p* = 0.1360)	−0.18 (*p* = 0.0689)	−0.30 (*p* = 0.0742)	0.04 (*p* = 0.8061)
Walking	0.15 (*p* = 0.1270)	0.14 (*p* = 0.1462)	−0.32 (*p* = 0.0508)	0.03 (*p* = 0.8574)
Total effort	0.01 (*p* = 0.9195)	0.02 (*p* = 0.8352)	−0.34 (*p* = 0.0405 *)	0.05 (*p* = 0.7673)

Spearman’s correlation coefficient with assessment of statistical significance (*p* value in brackets), statistically significant correlations were denoted with * for *p* < 0.05; IPAQ (International Physical Activity Questionnaire).

**Table 5 ijerph-18-06718-t005:** Correlations between the level of Internet addiction and back pain.

Spinal Pain	Sex	
	Female	Male
	A Measure of Internet Addiction	
NDI	0.20 (*p* = 0.0429 *)	0.37 (*p* = 0.0254 *)
ODI	0.19 (*p* = 0.0589)	0.10 (*p* = 0.5776)

Spearman’s correlation coefficient with assessment of statistical significance (*p* value in brackets), statistically significant correlations were denoted with * for *p* < 0.05.

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
