# Peer review of "Chronic Back Condition and the Level of Physical Activity as Well as Internet Addiction among Physiotherapy Students during the COVID-19 Pandemic in Poland"

_ijerph, 2021, doi:10.3390/ijerph18136718_

Round 1

Reviewer 1 Report

Comments on the manuscript “Chronic back condition and the level of physical activity as well as Internet addiction among students during the Covid-19 pandemic in Poland”

Introduction:

  • It is not clear why authors address Internet addiction in the study.
  • The relationship between intensive Internet use and physical pain is mentioned in the first paragraph, but it is not clear why Internet addiction was measured.
  • Authors do not explain why the study included university students and physiotherapy students and not another population group.

Methods

  • The sample is said to be representative of physiotherapy students in Poland, but it is not clear what procedure was used to select this sample and invite them to answer the online questionnaire. Please describe what universities were selected. Furthermore, the calculation of the sample size is not presented.
  • Mentioning the inclusion criteria is enough; the exclusion criteria negate the inclusion criteria. Incomplete questionnaires must be considered criterion for removal and not for selection, because students were over 18 years of age and were physiotherapy students and gave their consent and started filling in the questionnaire. Then, questionnaires were reviewed and if they were incomplete, then they must be removed.
  • Please explain why your study is considered a self-controlled cohort study, since it is not clear why nor your study has the characteristics of a cohort study.
  • Please mention how inferential statistical methods were used, because it is not clear what they were used for.

Results

  • When describing the results, the numbers of the tables nor the figures they represent are not mentioned.
  • In table 3 what do the double asterisk (**) and triple asterisk (***) mean? Please explain this.
  • In table 4 you must specify the meaning of both an asterisk (*) and the color shading in the columns.
  • In figure 2, what do “kobieta” and “mḛzcyẑna”mean? On the x-axis, the title is in Polish.

Discussion:

  • In the last paragraph, the small sample size is mentioned as a limitation of the study, so authors should include the calculation of the sample size.

Author Response

Response to Reviewers

Reviewer #1

Introduction:

It is not clear why authors address Internet addiction in the study.

The relationship between intensive Internet use and physical pain is mentioned in the first paragraph, but it is not clear why Internet addiction was measured.

Authors do not explain why the study included university students and physiotherapy students and not another population group.

Thank you for your remark. The authors assumed that Internet addiction is connected with its intensive use which was observed during online learning in the course of the pandemic .         It should be remembered that long-term use of the Internet is associated with a forced body position. The selected test group is able to name the pain problem that occurs while using the Internet. The research covered students of physiotherapy (after the corrections it was also indicated in the title that the subjects were students of physiotherapy). The group was selected due to the practical profile of education and the large number of activities involving physical activity that were not possible during the pandemic..

Methods

The sample is said to be representative of physiotherapy students in Poland, but it is not clear what procedure was used to select this sample and invite them to answer the online questionnaire. Please describe what universities were selected. Furthermore, the calculation of the sample size is not presented.

Thank you for your remarks. The text has been corrected. From the collected questionnaires, every fifth person was randomly selected from each year of studies The following universities in Poland have been selected: Lomza State University from Applied Sciences, School of Medical Sciences in Białystok, Medical University of Bialystok, Medical University from Warsaw

The article does not include sample size calculations, because the subject of our considerations was not one size, but several complex measures: ODI, NDI, IPAQ. Sample calculation procedures can be meaningfully applied when the subject of consideration is one numerical variable or the frequency of certain events occurrence. In the case of testing several measures coming from several standardized questionnaires, it was difficult to clearly establish the minimum statistical error criterion. Therefore, the authors tried to make the largest possible questionnaire survey, as far as possible, of course. It should be emphasized that        the size of the sample clearly affects the results of the statistical tests carried out, but these are authors who bear the risk of not obtaining significant results due to too small sample size.  . However, it cannot be said that too small sample size depreciates the results of the research, as long as appropriate statistical tests are used.

Mentioning the inclusion criteria is enough; the exclusion criteria negate the inclusion criteria. Incomplete questionnaires must be considered criterion for removal and not for selection, because students were over 18 years of age and were physiotherapy students and gave their consent and started filling in the questionnaire. Then, questionnaires were reviewed and if they were incomplete, then they must be removed.

Thank you for your remarks. The text has been corrected.

Please explain why your study is considered a self-controlled cohort study, since it is not clear why nor your study has the characteristics of a cohort study. Please mention how inferential statistical methods were used, because it is not clear what they were used for.

Thank you for your comments, corrections have been made. In the text there was an error of the authors it was supposed to be a cross sectional study. It is true that it was not specified in detail in 2.3. for which comparisons the Mann-Whitney test or the Spearman correlation coefficient was used, but this information is under each table, so according to the authors, the reader will know what type of analysis was used.

Results

When describing the results, the numbers of the tables nor the figures they represent are not mentioned.

Thank you  for your remarks  The text has been corrected..

In table 3 what do the double asterisk (**) and triple asterisk (***) mean? Please explain this.

Thank you for your remarks The text has been corrected.

In table 4 you must specify the meaning of both an asterisk (*) and the color shading in the columns.

Thank you for your comments, the description has been applied. To facilitate the interpretation of the results, relatively stronger positive correlations are distinguished by shades of red, and negative correlations - of blue.

In figure 2, what do kobieta” and mḛzcyẑna”mean? On the x-axis, the title is in Polish.

Thank you for your comments. The text has been corrected.

Discussion:

In the last paragraph, the small sample size is mentioned as a limitation of the study, so authors should include the calculation of the sample size.

The article does not include sample size calculations, because the subject of our considerations was not one size, but several complex measures: ODI, NDI, IPAQ. Sample calculation procedures can be meaningfully applied when the subject of consideration is one numerical variable or the frequency of certain events occurrence. In the case of testing several measures coming from several standardized questionnaires, it was difficult to clearly establish the minimum statistical error criterion. Therefore, the authors tried to make the largest possible questionnaire survey, as far as possible, of course. It should be emphasized that        the size of the sample clearly affects the results of the statistical tests carried out, but these are authors who bear the risk of not obtaining significant results due to too small sample size.  . However, it cannot be said that too small sample size depreciates the results of the research, as long as appropriate statistical tests are used.

.

Reviewer 2 Report

Figures 1 and 2, the description is in polish language – please change this. Statistically significant???

In any description of table and figure, you cannot use short cast as, for example NDI, without explaining it.

For example, descriptions of Table 4 should in one part Correlations between back pain (ODI and NDI - explain) and the level of physical activity (IPAQ - explain). Spearman’s correlation coefficient with an assessment of statistical significance (p-value in brackets). Statistically significant differences were denoted with for p <0.05.

Page 4 “ On the basis of the standards created for the IPAQ questionnaire, the respondents were categorized into three groups of low (40.4%), medium (39.7%) and high (19.9%) physical activity” this should be in material and methods not in the result section.

Page 5 “Due to the high influence of gender on the analyzed measures (especially activity measures), the analyzes were performed with the subdivision into the groups of women and men” as above

“The Spearman's rank correlation coefficient with the assessment of statistical significance was shown in the tables. The table shows the Spearman's rank correlation coefficients between the severity of pain and physical activity (IPAQ measures)”. This should be  in section “Statistical methods.”

Page 6 “ Sitting in front of a computer or smartphone influences back pain. Such a relationship occurs mainly as for cervical pain (correlations are statistically significant for both sexes, but the correlation strength is greater for men: rS = 0.37 vs rS = 0.20), and in the case of female students, on the border of statistical significance there is the relationship between the Internet addiction and pain in the lumbosacral region (rS = 0.19; p = 0.0589).” this is not result!

In the discussion section should also be discussed part in which Authors will answer the question:

To what extent was the fact that the students of physiotherapy took part in the study influenced the research results, and to what extent, by succumbing to the subject of the study, they could unconsciously answer questions biased.

Author Response

RESPONSE TO REVIEWER

Reviewer #2

Figures 1 and 2, the description is in polish language – please change this. Statistically significant.

Thanks for your comments. The text has been corrected.

In any description of table and figure, you cannot use short cast as, for example NDI, without explaining it.

Thanks for your remarks. The text has been corrected.

Page 4 On the basis of the standards created for the IPAQ questionnaire, the respondents were categorized into three groups of low (40.4%), medium (39.7%) and high (19.9%) physical activity” this should be in material and methods not in the result section.

Thanks for your comments. The text has been corrected.

Page 5 Due to the high influence of gender on the analyzed measures (especially activity measures), the analyzes were performed with the subdivision into the groups of women and men” as above

Thanks for your remarks. The text has been corrected..

The Spearman's rank correlation coefficient with the assessment of statistical significance was shown in the tables. The table shows the Spearman's rank correlation coefficients between the severity of pain and physical activity (IPAQ measures)”. This should be  in section Statistical methods.

Thank you for your remarks. The authors understand that, in the Reviewer's opinion, such a sentence should be included in the Statistical methods section. However, the authors would like to note that the table presents the results of the analyzes.

Page 6 Sitting in front of a computer or smartphone influences back pain. Such a relationship occurs mainly as for cervical pain (correlations are statistically significant for both sexes, but the correlation strength is greater for men: rS = 0.37 vs rS = 0.20), and in the case of female students, on the border of statistical significance there is the relationship between the Internet addiction and pain in the lumbosacral region (rS = 0.19; p = 0.0589).” this is not result!

Thank you for your remark. The authors understand that, in the opinion of the Reviewer, such a sentence should be included in the Discussion section , and in the Result part there should be only information. However, the authors would like to note that the table presents the results of the analysis etc.

In the discussion section should also be discussed part in which Authors will answer the question:

To what extent was the fact that the students of physiotherapy took part in the study influenced the research results, and to what extent, by succumbing to the subject of the study, they could unconsciously answer questions biased.

Thank you for your comments. In the opinion of the authors of the study, it does not matter, because research will be carried out on other groups in the future, which may be one of the threads in the continuation of this publication.

Reviewer 3 Report

This is an interesting cross sectional survey based study.

The authors can definitely improve the quality of the manuscript with the help of a scientific writer for a better organization and succint way of presentation.

The results of the study are some what important to the public.

Author Response

RESPONSE TO REVIEWER

Reviewer #3

Thank you for your favorable review.

Round 2

Reviewer 1 Report

Please correct tab. by Table and fig. by Figure

Author Response

Thank you for your remarks. The text has been corrected.

Reviewer 2 Report

Still is missing an answer to my question

To what extent was the fact that the students of physiotherapy took part in the study influenced the research results, and to what extent, by succumbing to the subject of the study, they could unconsciously answer questions biased.

In the other words, they as professional staff could unconsciously answer in the way in which they were expected.

Author Response

Thank you for your remark. We would like to point out that the study group in the form of physiotherapy students was selected on purpose. On the one hand, students are victims of a pandemic like the entire population, on the other hand, they are people with sufficient knowledge and experience to identify the problem. The verified control of the questionnaires sent by students confirmed our belief that the students correctly identified the problems included in the questionnaires and correctly assessed the reasons for the problem, which was consistent with the theses assumed in the study. The authors believe that the participation of physiotherapy students did not affect the results, as similar studies carried out on a group of students of technical faculties gave comparable results.